# High-throughput atomistic modeling of nanocrystalline structure and mechanics of calcium aluminate silicate hydrate

Yunjian Li [1] ✉, Cheng Chen [1], Zhenning Li[2] & Zongjin Li [1]

Although aluminum-containing cements have gained attention as environmentally friendly construction materials, the nanocrystalline structure and mechanical behavior of their primary hydration product, calcium aluminate silicate hydrate (C-A-S-H), remain poorly understood due to its complex chemical composition and structural disorder. Here, we present a high-throughput atomistic modeling framework to systematically investigate the structural and mechanical properties of C-A-S-H across a broad range of Ca/Si (1.3–1.9) and Al/Si (0–0.15) ratios. The compositional, structural, and mechanical features of C-A-S-H are accurately captured by molecular dynamics simulations of 1600 distinct C-A-S-H structures constructed using our in-house automatic structure generation program, CASHgen. Our findings highlight the influence of Ca/Si and Al/Si ratios on key C-A-S-H characteristics, including the mean chain length (MCL), interlayer spacing, coordination number and elastic moduli. Specifically, C-A-S-H exhibits optimal mechanical performance at a Ca/Si ratio of approximately 1.5, while further increases in Ca/Si introduce disorder and reduce stiffness. In contrast, increasing the Al/Si ratio promotes chain polymerization, leading to longer MCLs and improved mechanical performance. These results provide atomic-scale insights into the structure-property relationships in C-A-S-H and offer design guidelines for high-performance, low-carbon cementitious materials.

As the most extensively used construction materials on the planet, cementitious materials are the cornerstone of modern infrastructure and civilization. However, the cement industry is a major contributor to global greenhouse gas emissions, responsible for approximately 8% of total $CO_2$ emissions worldwide[1,2]. Improving the sustainability of concrete to minimise waste, embodied energy, and $CO_2$ emissions remains a critical challenge[3–5]. Common strategies for reducing the carbon footprint of cement production include improving the efficiency of cement plants, design of new cements that require less energy and partially substituting clinker with alternative materials, such as supplementary cementitious materials (SCMs)[6–10]. SCMs encompass industrial by-products including fly ash and blast furnace slag, as well as natural minerals like limestone and metakaolin. These materials are typically aluminum-rich, leading to calcium-aluminate-silicate-hydrate (C-A-S-H) as the predominant hydration product in SCM-based cementitious systems. C-A-S-H holds significant potential for carbon sequestration and the immobilization of harmful substances. While numerous studies have explored the properties and hydration kinetics of SCMs-based cementitious materials[11–15], atomic-level research nonetheless remains limited. Advancing the understanding of C-A-S-H structure and its mechanical properties at the molecular scale is critical to optimizing the performance of sustainable cementitious materials.

The structure of C-A-S-H closely resembles that of calcium-silicate-hydrate (C-S-H), the principal binding phase in ordinary

[1]Faculty of Innovation Engineering, Macau University of Science and Technology, Macao, China. [2]State Key Laboratory of Internet of Things for Smart City, University of Macau, Macao, China. ✉e-mail: liyunjian@must.edu.mo

Portland cement, with its distinctive characteristic being the incorporation of aluminum into the molecular framework. Over the past years, researchers have developed detailed atomic models for C-S-H structures based on the atomic structure of 14 Å tobermorite[16–20]. The silicate units in this model are classified by $Q^1$, $Q^{2p}$, and $Q^{2b}$, and follows the 3n-1 Dreierketten arrangement[16,21,22]. By accounting for the missing bridging silicate tetrahedra and the interlayer species, key characteristics such as Ca/Si ratio, water/Si ratio, and MCL in C-S-H structures can be tailored to align with the experimental observation[23,24]. Nevertheless, the presence of random defects and the high variability inherent in the C-S-H structure pose significant challenges for atomic modeling, particularly in capturing its long-range disordered and short-range ordered nature. The block model proposed by ref. 20 brought versatility to the study of C-S-H defects by enabling the rapid identification and analysis of structural irregularities. Building on this foundation, the pyCSH code proposed by ref. 25 further advanced the large-scale, randomized construction of C-S-H structures. By allowing the customization of parameters such as the Ca/Si ratio, the code can generate numerous atomic structures with random defects.

For instance, the dominance of $Q^1$ silicates at high Ca/Si ratios predicted by the code aligns closely with findings from $^{29}$Si nuclear magnetic resonance (NMR) studies[26]. This approach combines flexibility and precision in C-S-H structure design, enabling large-scale predictions and broadening the scope of atomic-level structural research.

The understanding of C-A-S-H structure lags behind that of C-S-H, largely due to the complexity of the aluminum's chemical environment and its positional variability. Dynamic nuclear polarization NMR experiments have revealed that aluminates in C-A-S-H structures adopt tetrahedral, pentagonal, and octahedral coordination forms, while also confirming the non-existence of the TAH phase (third aluminate hydrate)[27]. $^{27}$Al NMR experiments further confirmed that tetra-coordinated aluminum atoms do not appear in the interlayer[28] (Fig. 1a). The aluminum tends to replace the calcium ions in the

interlayer and silicate tetrahedra at bridging, pairing, and ending sites along the chains[29] (Fig. 1a). Density Functional Theory (DFT) calculations pointed that the Al tends to replace the Si at bridging site than that at pairing and ending site[29]. This was further verified by using a more complex model with both DFT calculations and molecular dynamics (MD) simulations[30,31]. Recently, Zhu et al. introduced aluminum atoms into the C-S-H block model, providing foundations for construction of C-A-S-H atomic structures[28].

Assessment of the validity of C-A-S-H atomic structures is nontrivial. The structural characteristics of the C-A-S-H, such as MCL, 2H/Si, are closely related to the Ca/Si and Al/Si ratios. Relying solely on Ca/(Si + Al) as a variable to analyze structural characteristics is inherently limited and fails to capture the nuanced changes in C-S-H structures resulting from aluminum incorporation. Therefore, research focusing on the interplay between Ca/Si and Al/Si characteristics becomes particularly critical. Moreover, accurately measuring interlayer spacing and atomic distances in samples with high Ca/Si ratios presents significant challenges, as techniques such as X-ray diffraction (XRD) struggle to precisely measure the interlayer spacing under these conditions[28]. While most studies suggested that the interlayer spacing decreases as the Ca/Si ratio increases, this trend is generally observed within a narrower range of Ca/Si value around 1.5[32], which is lower than the typical Ca/Si ratio of 1.70 found in Portland cement.

Here, we conducted high-throughput atomistic simulations to investigate the nanocrystalline structure and mechanical behavior of C-A-S-H. To enable systematic exploration, we developed CASHgen, a custom structure generation program designed for the automatic construction of C-A-S-H atomic models. The program can generate C-A-S-H structures with a wide range of defect variations and structural properties, including Ca/Si ratio, Al/Si ratio, charge distribution, MCL, and aluminum coordination proportion. Using CASHgen, we generated 1600 distinct C-A-S-H structures covering Ca/Si ratios of 1.3, 1.5, 1.7, and 1.9, and Al/Si ratios of 0.05, 0.10, and 0.15 (with interlayer aluminum fixed at 0.15). The generated structures were validated

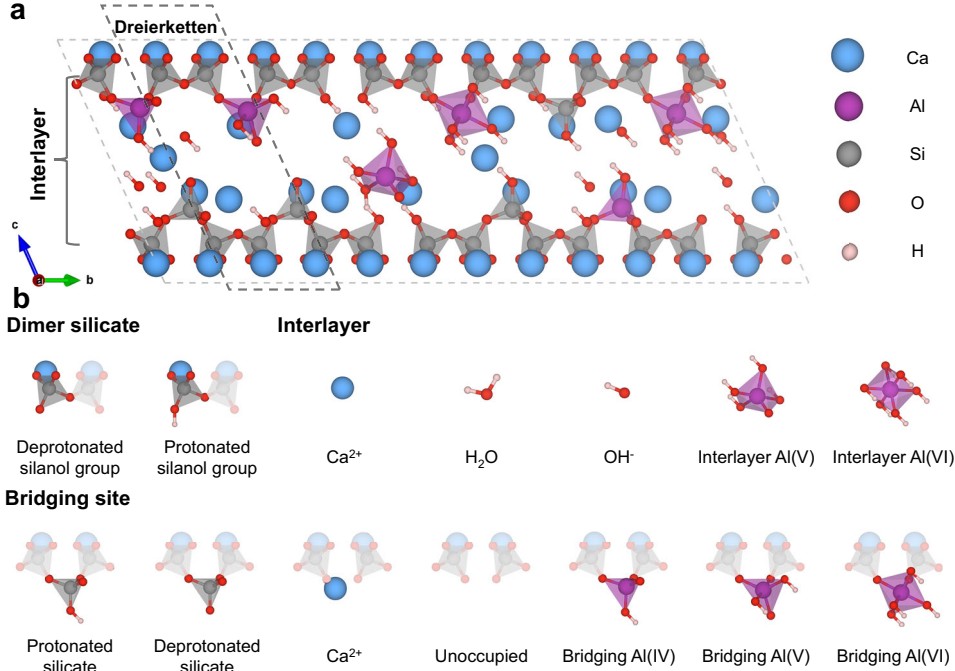

**Fig. 1 | Schematic representation of the supercell structure and atomic groups of C-A-S-H. a** C-A-S-H supercell. Water molecules in the interlayer are hidden for clarity, and the structures are unrelaxed. The black dashed box represents a segment of the C-A-S-H chain that is arranged in accordance with the dreierketten sequence, and the grey dashed box shows the boundary of the C-A-S-H supercell. **b** Atomic groups (shown as opaque) used to generate representative C-A-S-H structures. Transparent atomic groups highlight the connectivity of the atom groups within the silicate chains, which are adapted from the ref. 25.

against experimental data for key characteristics, such as MCL, 2H/Si, Ca-OH/Ca, Si-OH/Si. Through MD simulations, we further investigated the interlayer spacing, density, radial distribution function (RDF), and pair distribution function (PDF) of the C-A-S-H structures. Through systematic analysis of the equilibrated structures, we identified composition-dependent trends, particularly under high Ca/Si and Al/Si conditions, elucidating how atomic-scale changes influence overall structure and mechanical behavior. This work not only advances the understanding of C-A-S-H nanostructures, but also paves the way for the construction of a genetic database for C-A-S-H nanomaterials and bottom-up optimization of their performance.

## Results

### Description of C-A-S-H atomic groups

The atomic groups or 'genes', used for constructing C-A-S-H atomic structures are shown in Fig. 1b. At the pairing sites of the silicate chains, the structural units consist of pairing silicate dimers ($Q^{2p}$), while bridging sites may be occupied by bridging silicate dimers ($Q^{2b}$), bridging aluminate dimers ($Q^{2b}$ (Al)), or remain vacant or occupied by $Ca^{2+}$ ions. The $Q^{2b}$ can feature either two non-protonated oxygen atoms or one protonated oxygen atom. Aluminates at bridging sites exhibit diverse coordination states, including four-coordinated (Al(IV)), five-coordinated (Al(V)), and six-coordinated (Al(VI)) forms. According to experimental observations[28], bridging Al(V) typically carries two protonated oxygen atoms, whereas Al(IV) may exhibit one or two protonated oxygens depending on the local environment.

In the interlayer region, atomic groups include aqueous $Ca^{2+}$ ions, $H_2O$ molecules, $OH^-$ ions, as well as five-coordinated and six-coordinated aluminates (interlayer Al(V) and interlayer Al(VI)). The inclusion of interlayer aluminates is essential for modeling C-A-S-H with high Ca/Si ratios, especially under aluminum-rich conditions. The absence of Al(IV) in the interlayer is based on the $^{27}$Al NMR measurements[28,33]. Besides, interlayer Al(V) is modeled with four protonated oxygen atoms, while Al(VI) is treated as fully protonated, consistent with the $^{27}$Al [$^1$H] CP/MAS NMR results[34].

### Compositional characterization of C-A-S-H atomic structures

As the coordination states of aluminum and their relative proportions vary systematically with the Ca/Si ratio in C-A-S-H, it is essential to determine their accurate assignment when constructing realistic C-A-S-H models. Mohamed et al.[27] reported that at high Ca/Si ratios, Al(VI) dominates among aluminum coordination environments, while Al(IV) becomes the least favorable. Conversely, at low Ca/Si ratios, Al(IV) is the most prevalent, Al(VI) is minimal, and Al(V) consistently falls in between. Lothenbach et al.[35] compiled data from ref. 33 (Al/Si = 0.19) and Renaudin et al.[36] (Al/Si = 0.1) on the aluminum coordination number as a function of the Ca/Si ratio. Their findings indicate that the fraction of Al(V) remains stable at approximately 10%, regardless of the Al/Si ratio. Meanwhile, the fraction of Al(IV) decreases and Al(VI) increases as the Ca/Si ratio rises. To reflect this compositional behavior, we adopted the coordination ratios obtained by ref. 27 after structural equilibration at Ca/Si = 1.7, where the distribution of Al(IV), Al(V), and Al(VI) was determined to be 0%, 25%, and 75%, respectively. The corresponding aluminum coordination states used for selected Ca/Si ratios are summarized in Table S14.

Composition of the C-A-S-H models were analyzed based on 1600 structures with Ca/Si ratios ranging from 1.3 to 1.9 and from 0.05 to 0.15 with interlayer Al at 0.15. Based on existing experimental studies on the MCL of C-A-S-H and C-S-H across different Ca/Si and Al/Si ratios[26,28,37–52], a spatial surface diagram of MCL was fitted as a function of Ca/Si and Al/Si ratios, and corresponding cross-section plots with Al/Si ratios set at 0.05, 0.10, and 0.15 are shown in Fig. 2a, b. As most of the available experimental data fall within Al/Si ratios between 0 and 0.2, the surface model is constrained to the range of 0–0.15 for consistency.

We then compared the calculated MCL values from our atomic models to experimental results at Al/Si = 0.05, 0.10, and 0.15. These comparisons are presented in the cross-sectional plot of the fitted spatial surface diagram derived from experimentally measured MCL data (Fig. 2c–f). Both computational and experimental MCL present a decreasing trend with increasing Ca/Si ratios and increasing trend with the increase of Al/Si ratios. This reflects the role of aluminum in promoting silicate chain polymerization via aluminate bridging. Besides, the mean value of the MCL of C-A-S-H atomic structures generated by CASHgen closely matches the experimental data (Table S15). However, the standard deviation of the computational data increases with rising Ca/Si ratios. At low Ca/Si ratios, deviations from the ideal tobermorite structure (Ca/Si = 0.83) can be achieved in several ways, such as removal of bridging silicates or addition of interlayer calcium atoms. In contrast, at high Ca/Si ratios, $Ca^{2+}$ ions tend to occupy both interlayer and bridging sites in more consistent MCL values, thereby influencing the standard deviation. Similarly, the standard deviation also increases with higher Al/Si ratios, particularly as the system enters a high-aluminum state with Al/Si reaching 0.15 (Fig. 2e). Notably, when a fraction of bridging aluminates migrates from the silicate chains into the interlayer, this deviation is mitigated (Fig. 2f). This finding underscores the importance of incorporating interlayer aluminates under high-aluminum conditions, aligning with experimental observations[53].

The 2H/Si, Si-OH/Si, and Ca-OH/Ca ratios of the C-A-S-H atomic structures were calculated and compared with the experimental results, showing good agreement (Fig. 3a, b). For the 2H/Si ratio, it rises at a given Ca/Si ratio as the Al/Si ratio increases. The zoomed-in section in the lower-right corner illustrates that the inclusion of interlayer aluminates does not substantially affect the 2H/Si ratio at the same Al/Si ratio. This trend in 2H/Si is further supported by experimental validation. In the results of Yin et al.[6], the Al/Si ratio is highest at low Ca/Si, while at Ca/Si values approaching 1.8, the Al/Si ratio decreases to 0[6]. Comparing this findings with the results of Cong et al. reveals that for the same Ca/Si ratio, the 2H/Si ratio increases as Al/Si rises[54]. This trend is also consistent with the chemical composition statistics reported by Myers et al. for different Al/Si ratios[55]. For the Si-OH/Si ratio, it decreases as the Ca/Si ratio increases, which can be explained by the inhibition of the hydroxylation of silicates due to the occupation of the $Ca^{2+}$ ions on the bridging sites. While the Al/Si ratio has no significant effect on Si-OH/Si ratio. This can be explained by the inhibition of the hydroxylation of silicates due to the occupation of the $Ca^{2+}$ ions on the bridging sites. In contrast, the Ca-OH/Ca ratio increases with the Ca/Si ratio and decreases as the Al/Si ratio rises.

### Structural characterization of C-A-S-H atomic structures

The structure characteristics of the C-A-S-H models were further analyzed by molecular dynamics (MD) simulations. We randomly selected 160 samples with different Ca/Si and Al/Si ratios from the total generated 1600 structures. The reported values were averaged over 10 samples at specific Ca/Si and Al/Si ratios and calculated from the final 2 ns of the production run.

The interlayer spacings calculated from the C-A-S-H atomic structures with different Ca/Si and Al/Si ratios are shown in Fig. 3c. It increases with the rise in either the Ca/Si or Al/Si ratio, exhibiting a generally linear relationship. For example, at Al/Si = 0.1, the interlayer distance increases from Ca/Si = 1.3 (12.76 Å) to Ca/Si = 1.9 (16.10 Å), maintaining a linear growth pattern. While an increase in Al/Si leads to a slight increase in the interlayer spacings. When Al/Si = 0.15, introducing a small amount of aluminum into the interlayer results in a slight decrease in the interlayer distance, except for the case where Ca/Si = 1.3 (12.80 Å–12.85 Å).

As measuring interlayer spacings for C-A-S-H with high Ca/Si ratios is difficult[28], Renaudin[56] reported the C-A-S-H interlayer spacings at relative low Ca/Si ratio, which reveals a trend where the spacing initially decreases and then increases as the Ca/Si ratio rises (Fig. 3c). This

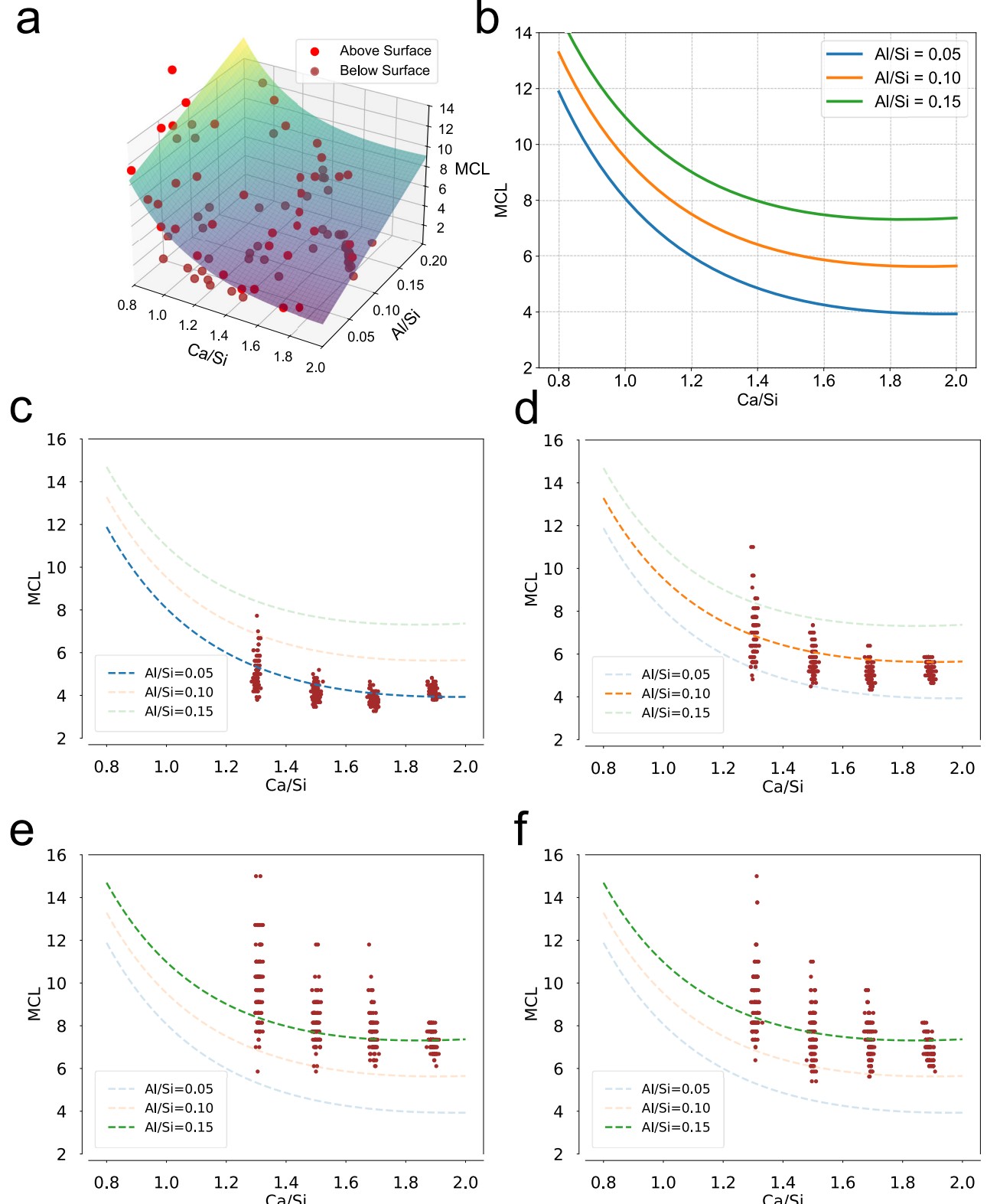

**Fig. 2 | Surface and cross-sectional plots of mean chain length (MCL) vs. Ca/Si and Al/Si ratios. a** Spatial surface diagram of MCL as a function of the Ca/Si and Al/Si ratios. The curved surface was constructed through the fitting of data points regarding the relationships between MCL and Ca/Si, as well as MCL and Al/Si, which were sourced from the literature[26,28,37–52]. **b** Cross-sectional plot of MCL as a function of the Ca/Si ratio, with Al/Si ratios set at 0.05, 0.10, and 0.15. **c** Al/Si = 0.05. **d** Al/Si = 0.10. **e** Al/Si = 0.15 without interlayer aluminates. **f** Al/Si = 0.15 with interlayer aluminates (refer to the Al(V)). The scattered points represent the MCL results for the C-A-S-H structures generated by the CASHgen program.

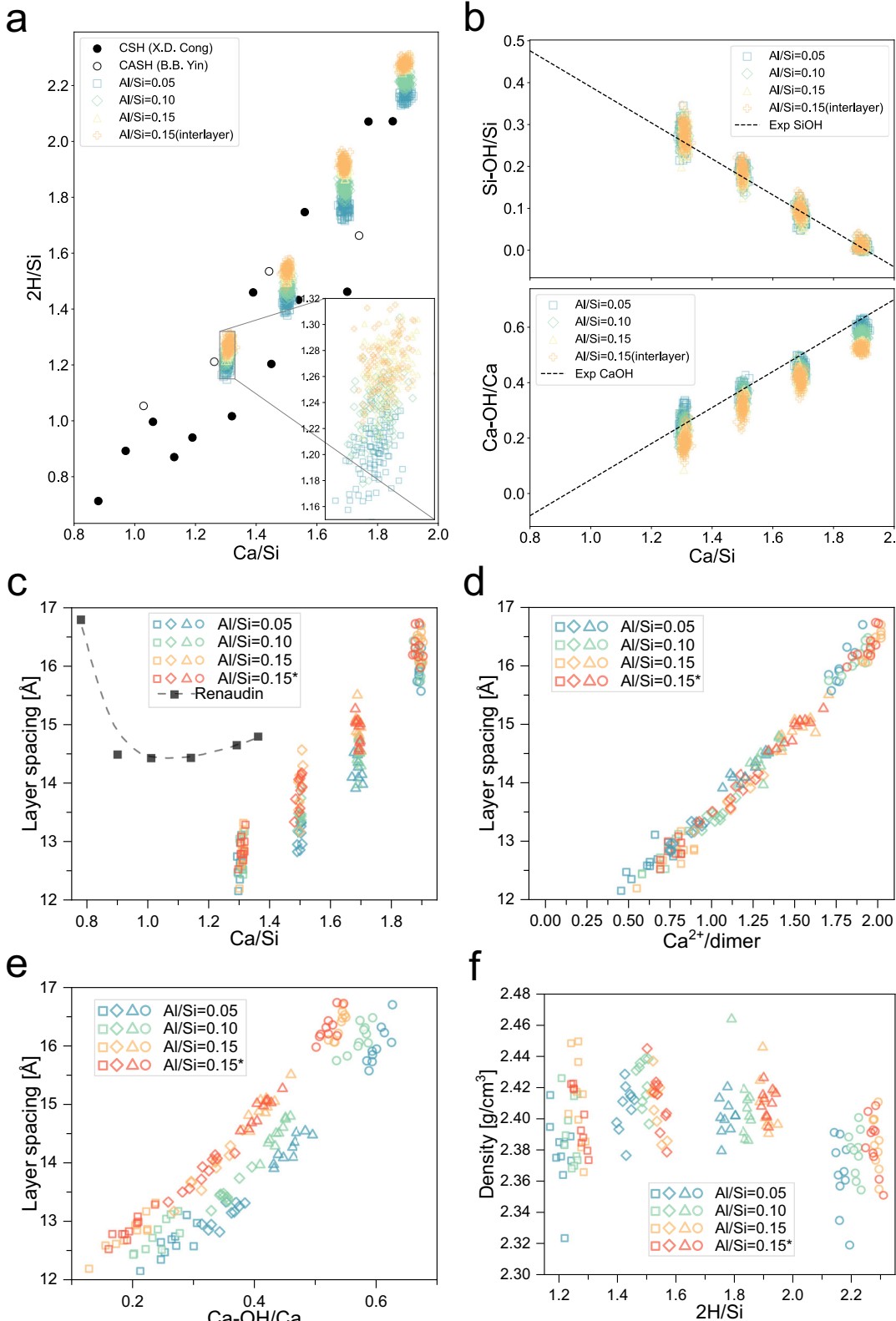

**Fig. 3 | Comparison of predicted composition and structural properties of C-A-S-H with experimental data. a** The 2H/Si ratios of the bulk C-A-S-H structures generated at the studied Ca/Si and Al/Si ratios are compared with the experimental data for C-A-S-H from ref. 6 and for C-S-H from ref. 54. **b** The Si-OH/Si and Ca-OH/Ca ratios of the bulk C-A-S-H structures generated at the studied Ca/Si and Al/Si ratios are compared with the experimental data from ref. 47. Layer spacings (z) as a function of **c** Ca/Si ratio, **d** Ca²⁺/dimer, and **e** Ca-OH/Ca. **f** Density as function of 2H/Si. The dot lines in (**c**) are experimental fitting curves from the ref. 56. '*' refers to the presence of interlayer Al(V) in the C-A-S-H structure. The square, diamond, triangle, and circle represent Ca/Si ratios of 1.3, 1.5, 1.7, and 1.9, respectively.

decrease at low Ca/Si ratios is attributed to the attraction between positively charged interlayer ions and adjacent negatively charged silicate chains. In contrast, this work focuses on a higher range of Ca/Si ratios (1.3–1.9), where the interlayer spacing exhibits an increasing trend with the increase of excessive interlayer $Ca^{2+}$ ions (Fig. 3d). This behavior is attributed to electrostatic repulsion between interlayer ions of the same charge.

Assuming that, at equilibrium, the interlayer $Ca^{2+}$ ions can occupy vacant bridging sites, the method involves first subtracting the number of vacancies from the $Ca^{2+}$ ions quantity and then dividing by the number of silicate dimers. This approach has been proven effective in evaluating the impact of $Ca^{2+}$ ions on the interlayer spacing. The results indicate that as the Ca/Si ratio increases, the $Ca^{2+}$/dimer ratio also increases (Fig. 3d). Figure 3e shows that as Ca-OH/Ca increases, the increase in interlayer spacing is also linearly correlated. At the same time, an increase in the Al/Si ratio promotes the expansion of the interlayer spacing. The interlayer spacing values are provided in Table S16, and it is evident that the spacing increases as the Ca/Si ratio rises.

The density of the C-A-S-H structure increases from Ca/Si = 1.3 to Ca/Si = 1.5 and then decreases (Fig. 3f and Table S16). When interlayer atoms are fewer, an increase in the Ca/Si ratio not only attracts the silicon chains of adjacent layers closer together, reducing the interlayer distance, but also increases the density. Conversely, the excessive addition of $Ca^{2+}$, $OH^-$ and water into the interlayer results in a less tightly bound and less ordered structure, leading to a decrease in density.

The calcium coordination number shows a slight increase with the rise in both Ca/Si and Al/Si ratios (Table S17). This is because that as the Ca/Si ratio increases, more $Ca^{2+}$ ions fill the interlayer and coordinate with $OH^-$ and water, which is consistent with the trend simulated using reactive force field[57]. The influence of the Al/Si ratio on calcium coordination can be explained by the randomness in the occupation of bridging positions by $Ca^{2+}$. An increase in the Al content leads to more bridging positions being occupied by aluminum atoms. This increases the probability of calcium atoms being distributed within the interlayer, resulting in a higher coordination number in the interlayer.

By initially defining the coordination states and the proportion of aluminum at different Ca/Si ratios (Table S14), the aluminum coordination number (CN(Al-O)) was calculated before and after the MD simulations, as shown in Table S17. The CN(Al-O) values for unrelaxed and relaxed structure are closely aligned, which reflects the validity of the initial definition of CN(Al-O). Notably, for Ca/Si = 1.3, the CN(Al-O) value in the production structure (4.90) is slightly higher than that in the unrelaxed structure (4.75). In contrast, for other Ca/Si ratios, the CN(Al-O) in the production structures is slightly smaller than their unrelaxed counterparts. Mohamed et al.[27] discovered that at high Ca/Si ratios, Al(VI) becomes dominant, which further suggests that the CN(Al-O) increases as the Ca/Si ratio rises. This also demonstrates that CASHgen is suitable for studying aluminum coordination issues.

The PDF in Fig. 4a and the RDF in Fig. 4b, c are obtained by averaging the results corresponding to the same Ca/Si and Al/Si ratios. A comparison between the individual PDFs and the mean PDFs is provided in the SI, showing negligible deviation. White et al.[58] reported experimentally that C-A-S-H and C-S-H exhibit very similar nano-structural features, with only negligible differences in their PDFs (Fig. 4a). This can be understood because aluminum in the C-A-S-H system exists as a guest atom, which has a relatively small impact on the PDF. In the third set of PDF results in Fig. 4a, where Ca/Si is fixed at 1.7 and the Al/Si ratio increases from 0.05 to 0.15, a small shift is observed between the first two peaks (Si–O at 1.6 Å and Ca–O at 2.39 Å), appearing at 1.8 Å, which corresponds to Al–O. The fourth set of results, where Al/Si is fixed at 0.15, shows the Al–O peak only under high Ca/Si conditions. Compared to the experimental PDFs reported by ref. 58, our predicted structures show a notable similarity and demonstrate a good agreement in terms of structural features, particularly between 1.2 and 5 Å. Separate PDFs of all Ca/Si and Al/Si ratios were summarized in Fig. S9. The atomic ordering is highly ordered, with the Al-O peak only being observable in environments with high Ca and Al ratios. We also compared the experimental and computational XRD results for C-A-S-H atomic structure (Fig. S15). This analysis presents a high degree of consistency between the C-A-S-H atomic

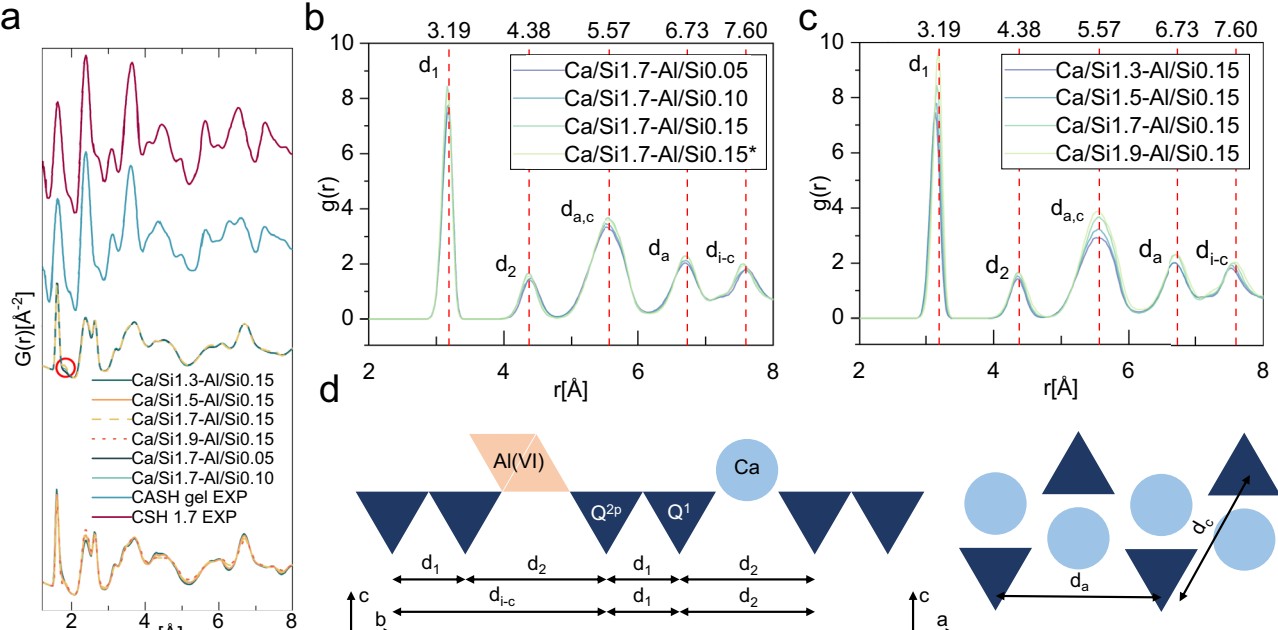

**Fig. 4 | Pair distribution functions and Si-Si radial distribution functions at varying Ca/Si and Al/Si ratios. a** Mean pair distribution function (PDF) of bulk C-A-S-H structure at the given Ca/Si or Al/Si ratios. The red circle highlights the Al-O peak at 1.8 Å. The experimental PDFs were sourced from ref. 58. **b** Mean Si-Si radial distribution function (RDF) at different Al/Si ratios (Ca/Si = 1.7); **c** Mean Si-Si RDF at different Ca/Si ratios (Al/Si = 0.15); **d** Annotation of the Si-Si distance. The dark blue triangle represents the silicate tetrahedron. In (**b**–**d**), the symbol 'd' refers to different types of Si-Si distances, and the red dashed lines are used to delineate the peak regions.

structure proposed herein and experimentally obtained XRD data, particularly in reflecting the characteristics of the C-A-S-H phase.

As silicates are the backbone of the C-A-S-H structure, analyzing the Si-Si distances can provide insights into the influence of the Ca/Si and Al/Si ratios on the atomic arrangement. Figure 4b, c present the changes in Si-Si distances caused by variations in the Al/Si and Ca/Si ratios. The first Si-Si distance (3.19 Å) corresponds to the spacing between Si atoms in adjacent silicates ($d_1$). The second Si-Si distance (4.38 Å) represents the distance between silicates separated by a bridging site ($d_2$). The fifth Si-Si distance (7.60 Å) corresponds to the length of a silicate chain that is further separated ($d_{i-c}$). As the aluminum content increases, both $d_1$ and $d_2$ show a slight decrease, while the reduction in $d_{i-c}$ is more noticeable. This can be viewed as the combined effect of $d_1$ and $d_2$, suggesting that the incorporation of aluminum enhances the connectivity within the silicate chains. In contrast, with the increase in the Ca/Si ratio (Fig. 4c), the silicate chains tend to elongate, mainly due to the slight upward shift of the first Si-Si distance ($d_1$). However, it is important to note that the increase in the Ca/Si ratio is a result of both calcium and silicate changes, so this phenomenon cannot be entirely attributed to the effect of $Ca^{2+}$ alone. The variations in the Ca/Si and Al/Si ratios do not have a significant effect on the movement of Si-Si distances along the a-axis and c-axis ($d_{ac}$ and $d_a$), indicating that the inter-chain distance along the a-axis and the intralayer distance along the c-axis in the C-A-S-H structures are relatively stable. Therefore, the analysis of interlayer spacing changes can be focused on the interlayer distance. As shown in the RDF plot (Fig. S10), the peak at 8.86 Å shifts to a greater distance with the increase in Ca/Si ratio, demonstrating that the change in interlayer spacing is primarily driven by the variation in the interlayer distance.

## Mechanical property analysis

The bulk modulus, Young's modulus, and shear modulus were first calculated based on the Voigt-Reuss-Hill (VRH) method[59–61] (Fig. 5a–c and SI section 5). These values reflect the trend in the overall mechanical properties of the C-A-S-H structure as a function of changes in Ca/Si and Al/Si ratios. C-A-S-H structure at higher Al/Si ratios exhibiting significantly higher moduli, meaning that the increase in Al/Si ratio has a strengthening effect on the mechanical properties. However, for the increase in Ca/Si, the module shows a trend of initially increasing and then decreasing. This contrasts with the trend reported by ref. 62, where the modulus decreases as Ca/Si increases. Therefore, further analysis of the anisotropy of the C-A-S-H structure is necessary to clarify the underlying causes of this variation.

The elastic tensor (Table S19), predicted using the stress-strain method, clearly demonstrates the anisotropy of C-A-S-H structures, with distinct variations depending on the Ca/Si and Al/Si ratios. Figure 5d–f show Young's modulus projection maps for the C-A-S-H structure with a fixed Al/Si ratio of 0.1, while Fig. 5g–i present the projection maps with a fixed Ca/Si ratio of 1.7. The Young's modulus projection maps for specific Ca/Si and Al/Si ratios are summarized in Figs. S17–S19. The impact of the Al/Si ratio on the Young's modulus of the structure can be observed in Fig. 5g. The increase in Al/Si enhances the Young's modulus along the Y direction, which can be explained by the occupation of bridging sites by Al atoms, forming Al-O-Si bonds that connect the silicon chains. In Fig. 5i, the effect of the Al/Si ratio on the structure in the X and Z directions is not significant. There is a very slight increase in the Young's modulus along the X direction, which is due to the shared oxygen atoms between a small number of Al and Ca. The stability of the Young's modulus in the Z direction can be attributed to the fact that only the Al = 0.15* group in the model has an Al atom entering the interlayer, and this effect is undoubtedly very weak. The influence of the Ca/Si ratio on the structure is clearly illustrated in Fig. 5d–f. As shown in Fig. 5d, with increasing Ca/Si, the Young's modulus in the XY plane consistently decreases. The reduction in the Y

direction is directly linked to the reduction in bridging silicon atoms, which causes chain breakage. Si-O bonds either disappear or are replaced by Ca-O bonds. This substitution also leads to the convergence of the Young's modulus in the X and Y directions at higher Ca/Si ratios. At high Ca/Si, silicon chains are primarily connected through Ca-O bonds, and as the Ca/Si ratio increases, the number of Ca-O linkages between the silicon chains also gradually increases. The variation in Young's modulus along the Z direction is the primary factor responsible for the observed increase in the overall Young's modulus of the C-A-S-H structure in Fig. 5b. When Ca/Si = 1.3, the Young's modulus is significantly smaller, and the Z-direction Young's modulus remains nearly the same for the other ratios (Ca/Si = 1.5, 1.7, 1.9). At the atomic level, it is speculated that when Ca/Si = 1.3, there are fewer Ca atoms in the interlayer, resulting in a relatively weaker connection between the upper and lower silicon chains. As shown in Fig. 3f and Table S16, the density for Ca/Si = 1.3 is lower than for Ca/Si = 1.5, indicating that the C-A-S-H structure has not yet reached a dense state. In contrast, at higher Ca/Si ratios, the Ca filling in the interlayer reaches a threshold, where the upper and lower silicon chains are tightly connected by an internal Ca layer in the interlayer.

To discuss the atomic-level explanation of the changes in Young's modulus in the three directions (X, Y, Z) with varying Ca/Si ratios, further validation can be obtained through the model parameters of the C-A-S-H structure. By analyzing the relationship between the three elastic tensors and the model parameters, the key factors responsible for the changes in Young's modulus can be identified. As shown in Fig. 5j–k, both $C_{11}$ and $C_{22}$ exhibit a linear regression relationship with MCL. This suggests that the breaking of Si-O bonds in the silicon chains is one of the key factors contributing to the decrease in Young's modulus along the X and Y directions. Moreover, the slope of the fitting line for $C_{22}$ (1.77) is larger than that for $C_{11}$ (1.25), indicating that the modulus in the Y direction is more sensitive to the breaking of silicon chains due to the increase in Ca/Si. This also confirms the previous analysis that the Young's modulus in the X and Y directions tends to converge as the Ca/Si ratio increases. The change in the modulus in the Z-direction must be attributed to the interlayer $Ca^{2+}$ ions, which plays a primary role. In the atomic simulations, the size variation of the simulation box is concentrated in the Z-direction, while the dimensions in the X and Y directions remain stable. As shown in Fig. 5l, the x-axis represents the linear density of interlayer $Ca^{2+}$, calculated by dividing the number of interlayer $Ca^{2+}$ ions by the interlayer spacing for each model. The y-axis corresponds to $C_{33}$. Here, structures containing interlayer Al atoms are excluded, and only the influence of $Ca^{2+}$ ions is considered. It is observed that when the linear density of interlayer $Ca^{2+}$ is low, $C_{33}$ is sensitive, but once it reaches a certain linear density, it stabilizes. This confirms the variation in the effect of interlayer $Ca^{2+}$ ions on the Young's modulus in the Z-direction.

## Discussion

C-A-S-H structure is a nanocrystalline material formed by the incorporation of the guest atom, aluminum into the C-S-H structure and their atomic arrangements are highly similar[58]. By analyzing the changes in structural parameters, we can predict the alterations in structural characteristics and performance, thereby enhancing our atomic-level understanding of the structure.

The MCL surface model derived from experimental data (Fig. 2a, b) is applicable to the structural analysis of C-A-S-H atomic structures with the Ca/Si ratio in the range of 1.3 to 1.9 and the Al/Si ratio from 0.05 to 0.15. Data for low Ca and high Al conditions are relatively scarce because aluminum tends to preferentially occupy bridging sites[47]. In low-calcium environments, there are fewer vacant bridging sites available for aluminum incorporation. Based on the automatically generated C-A-S-H structures, there is an interplay among the parameters determined during the initial generation (Ca/Si, Al/Si, MCL, Si-OH/Si, Ca-OH/Ca, charge).

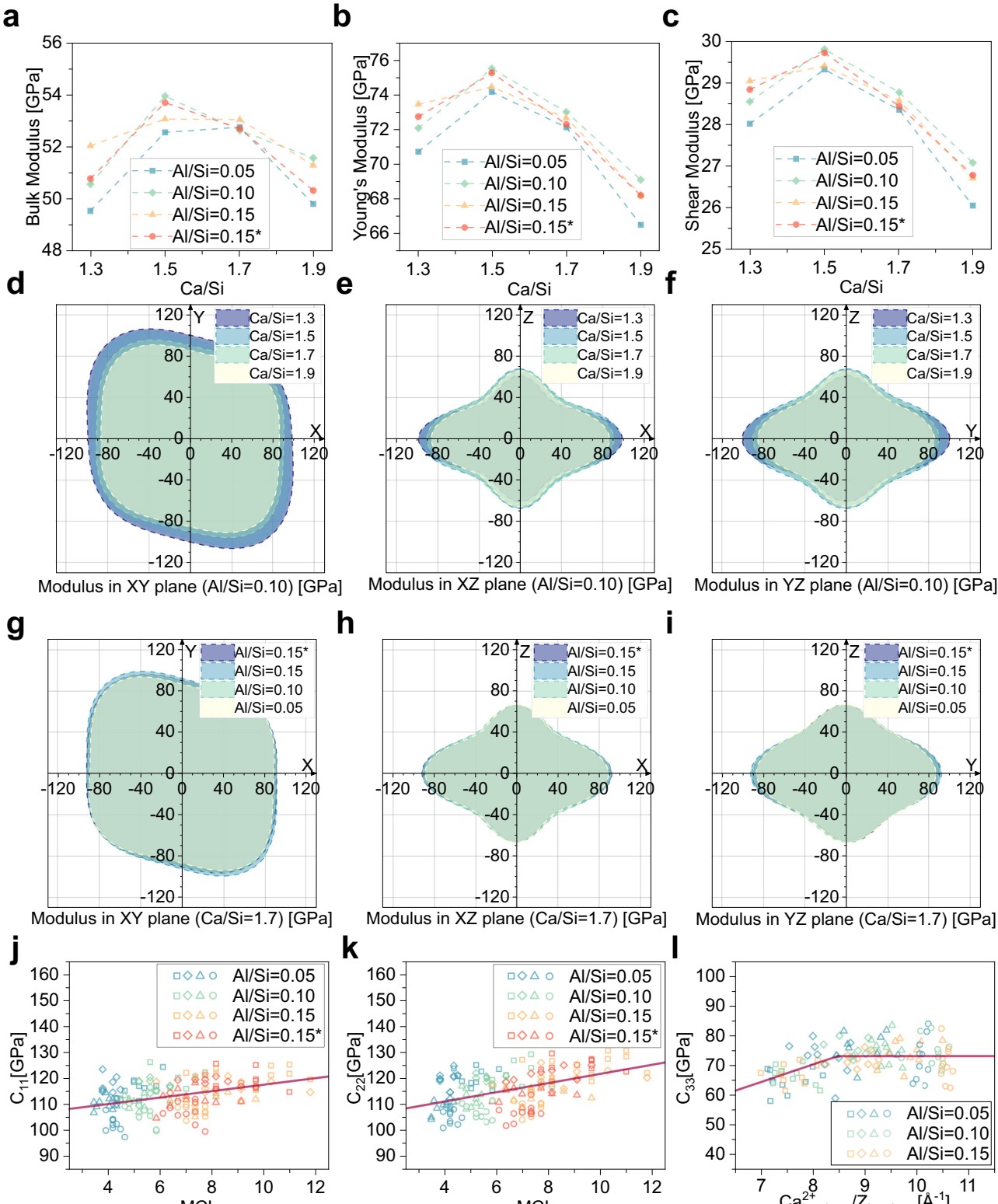

**Fig. 5 | Mechanical properties of C-A-S-H at varying Ca/Si and Al/Si ratios. a** Bulk modulus approximated by the Hill scheme. **b** Young's modulus approximated by the Hill scheme. **c** Shear modulus approximated by the Hill scheme. **d**–**f** Comparisons of the Young's modulus projections on the XY, XZ, and YZ planes for different Ca/Si ratios (with Al/Si = 0.10). **g**–**i** Comparisons of the Young's modulus projections on the XY, XZ, and YZ planes for different Al/Si ratios (with Ca/Si = 1.7). **j**, **k** Linear regression relationship between the elastic tensor components $C_{11}$, $C_{22}$, and Mean chain length (MCL). **l** Relationship between $C_{33}$ and the linear density of interlayer Ca. The first number in the subscript of the elastic constants represents the direction of the applied force, while the second number indicates the material's response in the corresponding direction. The red line is derived from the linear fitting. '*' refers to the presence of interlayer Al(V) in the C-A-S-H structure. The square, diamond, triangle, and circle represent Ca/Si ratios of 1.3, 1.5, 1.7, and 1.9, respectively.

Our results show that the Al/Si and Ca/Si ratios have a great influence on the MCL, 2H/Si, and Ca-OH/Ca ratios in C-A-S-H structures. The MCL exhibits a clear dependence on these ratios, which is influenced by the occupation of bridging sites in the silicate chains by aluminum or silicon. The increasing trend of 2H/Si for C-A-S-H structures with the increase of the Al/Si ratio, as shown in Fig. 3a, is attributed to the introduction of additional hydroxyl groups by aluminates compared to silicates. This process, involving interlayer aluminates absorption, also results in a decrease in the Ca-OH/Ca ratio, as observed in Fig. 3b, suggesting the competition between aluminates and calcium ions for OH groups. Theoretically, a higher Ca/Si ratio should introduce more pairing silicates due to the occupation of $Ca^{2+}$ ions at bridging sites[16,63]. However, an opposite scenario is observed, with the Si-OH/Si decreasing as the Ca/Si increases (Fig. 3b). We inferred that a charge balance mechanism governs this discrepancy. This accurate reflection of changes in atomic interactions in C-A-S-H atomic structures provides a foundation for the model's validity.

The variation in interlayer spacing is a complex issue, and the observed trend (Fig. 3c) does not align with many experimental reports[32]. However, the study by ref. 56. provided some insights. We established an analysis of the interlayer spacing in relation to $Ca^{2+}$ (Fig. 3d–f), which reveals a direct relationship between interlayer spacing and calcium content, particularly when calcium is present in the interlayer (Fig. 3d). This suggests that, in the C-A-S-H structure, as the Ca/Si ratio increases from low to the range between 1.3 and 1.9, the system has already passed the stage where interlayer spacing decreases with increasing Ca/Si ratio and has begun to transition into an expansion phase. The Si-Si peaks of the C-A-S-H structure indicate that changes in interlayer spacing are primarily caused by variations in the interlayer distance, rather than changes in the intralayer distance (Figs. 4b, c and S10). The presence of calcium and aluminum in the interlayer leads to the compression of the intralayer distance, causing the adjacent layers to move closer together and compress the interlayer distance until a tightly packed intralayer is formed. Conversely, excessive calcium or aluminum in the interlayer will cause the tightly packed interlayer to expand again. Based on the density variations of C-A-S-H, it can be inferred that the limit of this compression occurs between Ca/Si ratios of 1.3 and 1.5 (Table S16). Changes in interlayer spacing further affect the elastic modulus of the C-A-S-H structure. The linear density of $Ca^{2+}$ ions in the interlayer must reach a certain threshold before a stable elastic tensor $C_{33}$ appears (Fig. 5l). This growth phase can be interpreted as the process in which the interlayer begins to become more compact. Once a tightly packed interlayer is formed, further increases in the Ca/Si ratio do not significantly alter the elastic tensor of the C-A-S-H structure in the Z direction. This suggests that, for C-A-S-H, the elastic modulus in the Z direction is primarily governed by atomic interactions within the interlayer. In Fig. 5j–k, we also establish a relationship between MCL and the $C_{11}$, $C_{22}$ components of the elastic tensor. By analyzing the mechanical properties of a large number of structural models, we demonstrate that the performance of C-A-S-H can be directly predicted from fundamental parameters such as Ca/Si ratio, Al/Si ratio, MCL, and atomic composition. This capability significantly broadens the potential for large-scale structural analysis and materials design.

Although the empirical force field, ericaFF2, has demonstrated good adaptability in C-A-S-H structures[64], the model's results are still based on classical MD simulations. In the model with Ca/Si = 1.3, the unrelaxed interlayer distance (14 Å) and the equilibrated interlayer distance (12.761 Å) show a noticeable difference. In the empirical force field, this condensation process may be insufficient, and the atomic position changes are relatively limited. As a result, this leads to the observed trend in the variation of the elastic tensor $C_{33}$ in Fig. 5l. We are more hopeful for the development of higher-precision force fields for large-scale structural predictions, such as those based on machine learning potentials derived from DFT.

The CASHgen program developed in this work can provide feedback on atomic-level changes based on structural parameters. In the bulk C-A-S-H structure, changes in MCL, Si-OH/Si, and Ca-OH/Ca are linked to atomic-level reactions, such as the detachment of silicate groups on the silicon chain or the occupation of aluminum in silicate sites, and the attraction of hydroxyl groups by aluminum. Additionally, CASHgen uses Gaussian sampling to identify the proportions of primary atoms (calcium, silicon, oxygen, and hydrogen in C-A-S-H) and estimating the number of dopant atoms (aluminum) gives the automated program greater flexibility and broader applicability. This methodology can also be extended to a broader range of nanocrystalline materials with other types of atomic substitutions or incorporation of different dopant elements, helping researchers design and improve formulations of nanomaterials.

In summary, this study provides a comprehensive study on the nanocrystalline structures of C-A-S-H, focusing on atomic composition, structural parameters, and characteristics, as well as mechanical performance through high-throughput atomistic simulations and various atomic-level analyses. The atomic structures of C-A-S-H were constructed through developing the CASHgen program, which was specifically developed for the automatic generation of C-A-S-H atomic models with varying chemical compositions and defects. The generated C-A-S-H structures are validated to capture the key characteristics of experimentally obtained C-A-S-H, such as Ca/Si, Al/Si, 2H/Si, Si-OH/Ca, Ca-OH/Ca ratios, and MCL, and can accurately predict the trend in structural performance of C-A-S-H, such as the interlayer spacing, density and elastic tensor with the variation of the structural parameters. Analyses demonstrate that both the variation in interlayer spacing and the presence of interlayer Ca are strongly correlated, collectively influencing the Young's modulus in the Z direction. Meanwhile, the MCL has a significant impact on the X and Y directions. Increasing the Ca/Si ratio initially enhances interlayer connectivity and stiffness, but beyond a critical threshold, it shortens the MCL and reduces the elastic modulus. Conversely, Al incorporation promotes longer silicate chains and enhances elastic moduli. This study highlights the critical role of composition in governing the nanocrystalline structure and mechanics of C-A-S-H, providing valuable insights into the atomic-level behavior of cementitious C-A-S-H, paving the way for the construction of their materials genome database and bottom-up optimization of their performance.

## Methods

### Development of CASHgen program for automatically generation of atomic scale C-A-S-H structures

To generate realistic C-A-S-H atomic structures, the CASHgen program was developed based on the C-S-H building block and the aluminum-containing building block methodologies[20,28]. CASHgen serves as a reliable tool for constructing C-A-S-H atomic models that accommodate various chemical environments and defects. The process begins with the generation of building blocks, where atomic groups are assembled into a 14 Å tobermorite unit cell, incorporating various types of defects[16].

Figure 6 shows the operation flow chart for the CASHgen program. The use of CASHgen begins with the definition of key parameters, including the supercell size, Ca/Si ratio, Al/Si ratio, and $H_2O$/Si ratio. Then, the CASHgen assembles the building blocks from atomic groups, with the total number of blocks depending on the permissible charge deviation within the system. CASHgen automatically calculates the necessary number of Al atoms in the supercell to achieve the user-defined Al/Si ratio. The first task is to fill the supercell with blocks containing Al, continuously adding them until the total Al atom count is satisfied. During this process, the coordination and placement of Al atoms (such as in bridging sites or interlayer) are also considered. To address the complexity of aluminate protonation states while maintaining usability, CASHgen allows users to modify the protonation

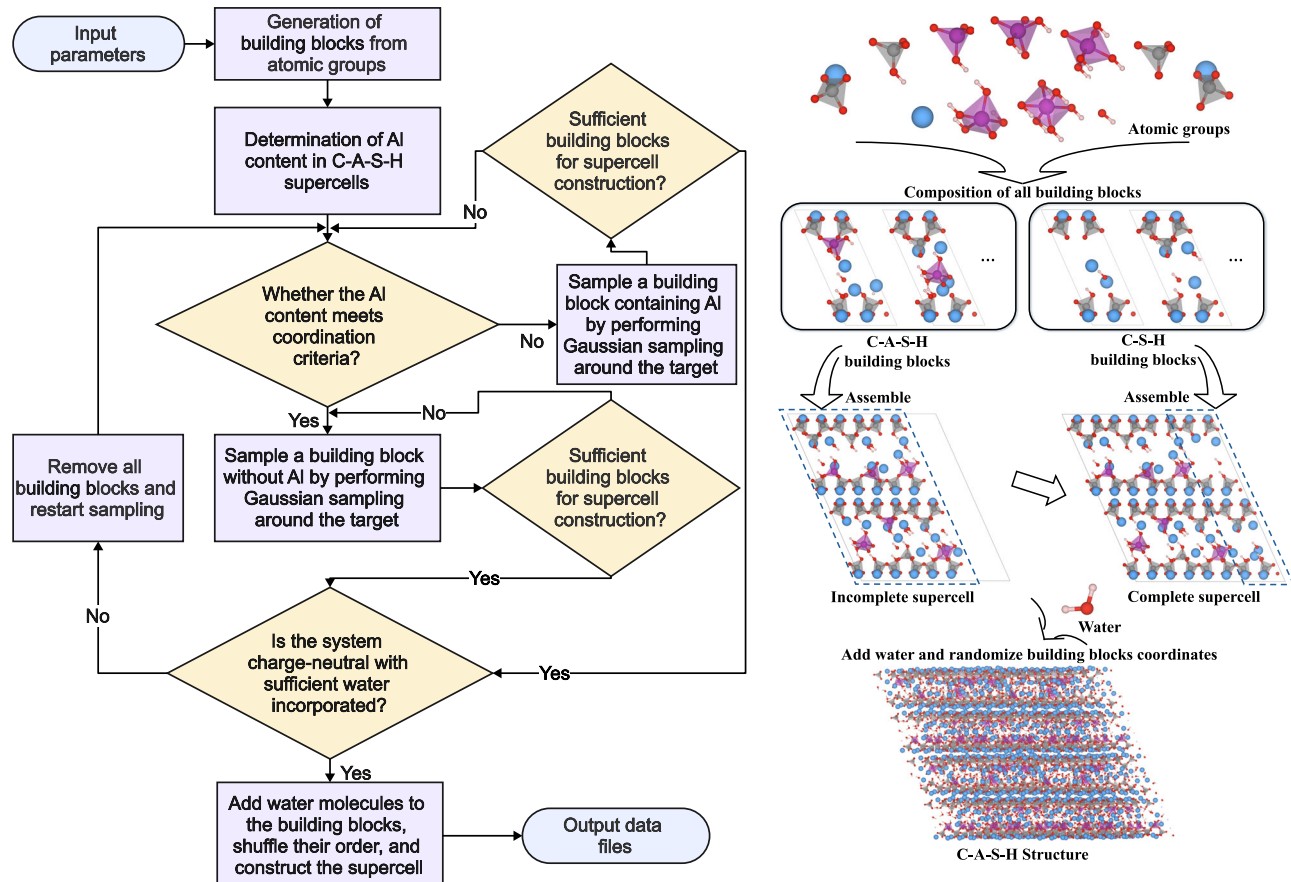

**Fig. 6 | Flowchart of the CASHgen program.** The flowchart outlines the steps for constructing a C-A-S-H supercell. First, building blocks are generated based on input parameters, and the required number of aluminum atoms is calculated. The program then checks if sufficient building blocks are available to build the super-cell. If not, new building blocks are generated using Gaussian sampling, ensuring compliance with Al content and coordination requirements. Next, the total charge is checked for neutrality, and the incorporation of sufficient water molecules is confirmed. If all criteria are met, water molecules are added, block coordinates are randomized, and the supercell is assembled and exported as data files. The right side of the figure illustrates this process. The right side of the figure visually depicts the composition and assembly process of different atomic blocks, including C-A-S-H and C-S-H building blocks. Water molecules are incorporated into these blocks to form the complete supercell structure.

states according to their needs. By default, it pre-defines one-proto-nated Al(IV), two-protonated Al(V), and four-protonated Al(VI).

Once all building blocks containing Al have been placed in the supercell, a preliminary supercell structure is formed. While not yet complete, it already meets the required conditions for Al atoms. The program then proceeds to fill the supercell with blocks without Al. All selected blocks are sampled using the Gaussian method to ensure that their parameters closely match the user-defined target values. CASH-gen calculates the deviation between the parameters of the blocks already placed in the supercell and the target values. Each time a new block is added, the target values are dynamically adjusted, improving the precision of the structure.

After these steps are completed, a dehydrated C-A-S-H structure is generated. If the charge balance is not achieved, all blocks are removed from the supercell, and the filling process is restarted. If charge bal-ance is attained, CASHgen randomly adds water molecules to the structure until the $H_2O$/Si ratio reaches the target value. This process is repeated, refining the placement of water molecules until it is as rea-sonable as possible.

A well-constructed C-A-S-H structure should have random defects to minimize human intervention. Therefore, the final step in CASHgen is to randomly reorder the unit cells within the supercell and present the user with the final C-A-S-H structure.

Based on the desired Ca/Si ratio, the CASHgen program adopts data from ref. 47. to obtain the corresponding Si-OH/Si and Ca-OH/Ca ratios. Additionally, data on C-A-S-H from ref. 6 has been incorporated into the 2H/Si ratio. For C-A-S-H structures with the same Ca/Si ratio, a consistent Water/Si ratio is defined, where "water" refers solely to water molecules, rather than the combination of water molecules and other hydrogen species. However, experimental methods typically assess the water content of the system using proton nuclear magnetic resonance ($^1$H NMR), making it challenging to differentiate the sources of hydrogen. Therefore, 2H/Si is effectively represented as $(H_2O + 2OH^-)$/Si. The average chain length of CASHgen is given by Eq. (1):

$$MCL = \frac{Q^1 + Q^{2b}(Si) + Q^{2b}(Al)}{0.5 \times Q^1 - Q^{2b}(Si) - Q^{2b}(Al)} \quad (1)$$

where $Q^{2b}(Si)$ represents the amount of bridging silicon, $Q^{2b}(Al)$ denotes the amount of bridging aluminum, and $Q^1$ refers to the amount of silicon in the paired site.

CASHgen adopts certain limitations when generating building blocks to ensure physical reliability. Firstly, silicates dimers containing Si-OH groups can only be selected if the bridging sites are either vacant

or occupied by $Ca^{2+}$ ions. This restriction is based on the silicon-oxygen linkage in silicates, which maintains the structural plausibility of the model. Additionally, when the interlayer aluminate is included, one bridging sites in the building block is set as vacant. However, under high-calcium conditions, it is still plausible for the bridging sites to be occupied by $Ca^{2+}$.

## Model construction

A total of 1600 initial supercells of C-A-S-H structures were generated for various Ca/Si ratios (1.3, 1.5, 1.7, and 1.9) and Al/Si ratios (0.05, 0.10, 0.15, with interlayer aluminum at 0.15), with 100 samples for each combination. Additionally, $^{27}$Al MAS NMR experimental results have revealed the presence of Al(V) in the interlayer[34,53]. Therefore, an additional set of C-A-S-H structures at Al/Si = 0.15 for various Ca/Si ratios, with 100 samples for each ratio were also constructed. These structures are particularly noteworthy because 20% of the aluminum atoms are in the form of interlayer Al(V), representing 5% of the total aluminum content.

## Molecular dynamics

In this work, all $4 \times 4 \times 2$ supercells of C-A-S-H were built by CASHgen and the atomic positions obtained by this method are predetermined, with the coordinates fixed. Three separate 40,000-step runs in the canonical (NVT) ensemble were performed. Energy minimization of the simulation box was then carried out to achieve a more favorable atomic arrangement with a lower relative energy. The force field used in this study is ericaFF2[64]. The core-shell model used in the force field can be referenced from the original study[65] and the LAMMPS documentation on core-shell models[66]. The structural heating and hard minimization process lasted for 1.65 ns with a timestep of 0.0002 ps. The final equilibration was performed under the NPT ensemble at 298.15 K and 1 atm for 800 ps.

The production run was conducted for 5 ns in the NPT ensemble at 298.15 K and 1 atm. Due to the effectiveness of the hard minimization procedure, the timestep for the production run was set to 0.0005 ps, with trajectory data recorded every 25 ps. Averages of temperature, pressure, energy, volume, and cell parameters were computed and recorded every 10 ps. The average potential energy of the core-shell pair was also calculated and recorded at the same interval. Throughout the production run, the bulk C-A-S-H system rapidly attained a stable state, with the energy remaining relatively constant over the course of the simulation. The c-axis of the simulation cell reached equilibrium approximately 2.5 ns into the run. Consequently, the average quantities reported in this study were derived from the final 2 ns of the simulation (Fig. S1).

The mechanical properties of bulk C-A-S-H were assessed using the stress-strain method, conducted at zero temperature. The time step for the simulation was set to 0.001 ps. The test structure was derived from the results of the equilibration and production run described earlier, which provided the starting configuration for the elastic constant calculations. Each simulation structure was run for 1.69 ns, comprising 13 cycles, with each cycle consisting of 100 ps of equilibration followed by 30 ps of production. Deformation magnitude of 2% were applied, and the resulting stress changes were used to compute the elastic stiffness tensor. For each model with a specific Ca/Si and Al/Si ratio, the elastic tensor values are averaged from the predictions of 10 models with the same ratio, which were shown Table S19. The mechanical properties, including the bulk modulus, shear modulus, Young's modulus, hardness, and Poisson's ratio were obtained by analyzing the stiffness matrix based on the VRH approximation[59–61], which were shown in Table S18.

All the simulations described above were conducted using LAMMPS (2 Aug 2023)[66]. The PDF and RDF were obtained by post-processing the trajectory files using the TRAVIS program[67,68]. The results were averaged over the entire 5 ns production run, and the RDF

for the atomic species were calculated with the exclusion of hydrogen atoms[69]. The PDF was subsequently calculated according to Eq. (2):

$$G(r) = \sum_{\alpha\beta} \frac{c_\alpha c_\beta b_\alpha b_\beta}{\langle b \rangle^2} G_{\alpha\beta}(r) \qquad (2)$$

where b represents the atomic scattering lengths of the atoms, and c is the weight factor corresponding to the atomic species fraction. The summations are taken over all atomic species in the sample. $G_{\alpha\beta}(r)$ represents the partial PDFs.

## Reporting summary

Further information on research design is available in the Nature Portfolio Reporting Summary linked to this article.

## Data availability

Relevant data is available in supplementary information and source data. 1600 C-A-S-H atomic structures generated by CASHgen, 160 equilibrated C-A-S-H atomic structures for mechanical properties analyses, along with data for PDF and RDF analyses, can be accessed on *figshare* at https://figshare.com/articles/dataset/Nanocrystalline_Structure_of_Calcium_Aluminate_Siliate_Hydrate/28696349[70]. Source data are provided with this paper.

## Code availability

The in-house automatic C-A-S-H atomic structure generation program, CASHgen, developed for this paper, is available for download at https://github.com/l98y3j26/CASHgen[71].

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

## Acknowledgements

The authors would like to acknowledge the Macao Science and Technology Development Fund (FDCT) (Grants No. 0074/2023/RIB3 [Y.L.] and 0139/2024/RIB2 [Y.L.]) and the Macau University of Science and Technology Faculty Research Grant (Grant No. FRG-24-085-FIE [Y.L.]). The authors sincerely thank Dr. Hegoi Manzano for the fruitful discussion on the manuscript, and thank the Beijing PARATERA Tech CO., Ltd. for providing resources for simulations.

## Author contributions

Y.L. conceived and supervised the project. Y.L. and C.C. designed the research. Y.L. and C.C. performed the simulations, analyzed the simulation results, and wrote the manuscript. Y.L., C.C., Z.N.L., and Z.J.L. discussed the results. Y.L. and C.C. edited the manuscript before submission.

## Competing interests

The authors declare no competing interests.
