## [Transparent Peer Review file · Nature Communications]

Nanocrystalline Structure of Calcium Aluminate Silicate Hydrate

Corresponding Author: Professor Yunjian Li

Version 0:

Reviewer comments:

Reviewer #1

(Remarks to the Author)

The manuscript presents the preparation of the CASH atomic structure using the proposed CASHgen program, with results reported through experimental comparisons and mechanical properties, among other aspects. The scope and scale of the work are substantial, and the authors should be commended for this. However, my primary concern pertains to the significance of the work. The current manuscript includes excessive modeling details in the main text, with some even being presented in the Results section. The authors should focus more on what insights can be derived from the constructed CASH models. The novelty and new findings should be clarified more clearly. Additionally, more thorough validation is required. As a result, substantial revisions are necessary before reconsideration for publication in Nature Communications.

Specific comments:

- The abstract should focus more on the new findings rather than on the modeling procedures and experimental verifications.
- The design of new cements that require less energy is also a common strategy for reducing the carbon footprint.
- The introduction is not logically structured. The authors attempt to include too much information, some of which is not directly related to the main objective of this study. For example, the main findings of this study do not directly address the hydration kinetics of SCMs-based cementitious materials or the underlying causes of prolonged setting times. It is unclear why these topics are included in the introduction.
- There are vague expressions throughout the manuscript. For instance, "but the combined influence of these factors remains challenging to evaluate." The combined influence of these factors on what?
- The authors should carefully review the content and determine what belongs in the Results section and what should be included in the Methods section. Excessive modeling details are currently presented in the Results section. Instead, the authors should focus on investigating the new findings and conclusions derived from the constructed CASH model.
- The validation of the proposed CASH model is insufficient. The current validation involves only a few previous studies and comparisons using several statistical indicators. The authors should synthesize representative CASH structures, conduct detailed experimental characterization (e.g., XRD, NMR), and provide a more comprehensive comparison.
- Section 2.3: Since the 160 samples were randomly selected, how can the authors ensure the results are representative? Please provide commentary on this.

(Remarks on code availability)

More detailed instructions about the code should be provided.

Reviewer #2

(Remarks to the Author)

The manuscript "Nanocrystalline Structure of Calcium Aluminate Silicate Hydrate" (NCOMMS-25-06753) addresses a critical issue in cement chemistry, offering a comprehensive and innovative approach to investigating the nanocrystalline nature of calcium aluminate silicate hydrate (CASH). The authors have developed a user-friendly code for the automatic construction of CASH atomic structures. The manuscript considers various potential defects in CASH structures and a wide range of Ca/Si and Al/Si ratios. The atomic models of CASH accurately predict key structural parameters and mechanical properties of CASH, aligning closely with experimental results. The findings hold significant real-world implications for aluminate-containing cements and other nanocrystalline materials. Notably, the work has the potential to contribute to the creation of a cement structure database, which could aid in the development of machine learning-based force field

potentials. Given the novelty, rigor, and potential impact of the study, I recommend publication in Nature Communications following the completion of the following minor revisions:

1. In Section 2.2, the authors summarize the results of some experimental work and perform a parameter surface fitting for the average chain length of CASH structures, based on experimental data collected at an Al/Si ratio below 0.2. Previous studies have typically considered the Ca/Si or Al/Si ratio as the sole variable affecting the average chain length. For CASH, it is difficult to accurately evaluate the influence of atomic content on chain formation or chain breaking. The authors' approach fills this gap and supports the accuracy of the structures generated by their proposed program. However, as high-alumina cements are increasingly recognized as green and sustainable materials, the author's fitting model currently only extends up to an Al/Si ratio of 0.2, with the applicability range stated as 0.15. Is there potential for further extension of this model?
2. In Section 2.3, the authors introduce a parameter, "Ca₂₊/dimer," which is related to the content in Figure 3(d). This parameter exhibits a strong linear relationship with the interlayer spacing. However, the authors do not provide a detailed explanation of "Ca₂₊/dimer." It is only briefly mentioned in the fourth paragraph of Section 2.3, where it is suggested to be related to the number of calcium atoms in the interlayer. A clearer and more comprehensive explanation of this parameter is needed in the manuscript.
3. The research in this manuscript is well-supported by detailed data, and the Results and Methods sections provide references to supplementary materials, for example, in the Method section: "(See SI section 3)." However, excessive reference to supplementary materials can make the reading and retrieval process more cumbersome. I recommend that the authors revise this by providing specific reference numbers, such as Figure S1, instead of general supplementary information.
4. Figure 4(d) lacks a legend for the Si data. I suggest that the authors add annotations to some of the triangles in the figure to avoid ambiguity.
5. In Figure 5(j-l), the authors correlate the parameters of the C-A-S-H structure with mechanical properties, which helps identify the key factors influencing mechanical performance. While previous studies have primarily focused on experimental analysis of the structures, simulations have often been limited by challenges such as supercell construction and computational costs. The approach presented in this manuscript, which analyzes large-scale data, fills a significant gap in the field. However, compared to the figure 5(j-k), the figure 5(l) lacks some structural data. Could the authors clarify whether this still accurately supports the conclusions drawn?
6. The link to the program at the end of Section 4.2 would be more appropriately placed under the Data/Code availability section, in line with standard formal requirements.

(Remarks on code availability)

Version 1:

Reviewer comments:

Reviewer #1

(Remarks to the Author)

The authors have adequately addressed most of my comments. I recommend the following minor revisions prior to acceptance in Nature Communications.

The discussion is quite straightforward—that is, somewhat superficial. A more in-depth analysis should explore the real-world implications of these results for the application of cementitious materials.

The abstract has been greatly improved; however, the findings section focuses primarily on the modulus. While the modulus is important, this emphasis appears too narrow and may lead readers to believe that the manuscript only investigates the relationship between various structural characteristics and the modulus. This perception is insufficient for a paper published in Nature Communications and could limit its appeal to a broader audience. Therefore, the abstract should be further refined.

(Remarks on code availability)

Instructions on code are clear and adequate.

Reviewer #2

(Remarks to the Author)

I would like to commend the authors for their diligent and thorough revision of the manuscript. The concerns and suggestions raised in the initial review have been comprehensively addressed, resulting in a significantly improved and polished manuscript.

The inclusion of more detailed discussions and the strengthening of the conclusions enhance the overall impact and rigor of the study. Furthermore, the manuscript now has a clear structure and logical flow, making the content accessible and engaging for readers. The figures are well-organized and effectively complement the text.

In light of these improvements, I am pleased to recommend this manuscript for publication in Nature Communications, as it presents novel insights that will greatly contribute to the field of cementitious materials genome database.

(Remarks on code availability)

Dear Reviewers,

We would like to sincerely thank you for your thorough and constructive comments on our manuscript titled “Nanocrystalline Structure of Calcium Aluminate Silicate Hydrate” (Manuscript ID: NCOMMS-25-06753). We greatly appreciate the time and effort you dedicated to reviewing our work. After carefully considering each of your comments, we have made the necessary revisions. Here are our **point-to-point responses** to the reviewers’ comments.

Reviewer #1

Comment 1: The abstract should focus more on the new findings rather than on the modeling procedures and experimental verifications.

Response: Thank you for your valuable feedback on our manuscript. We fully agree with your suggestion. In response to your comments, we have revised the abstract to emphasize the new findings and contributions of this study, while reducing the emphasis on the procedures and validation experiments. The revised abstract is as follows:

Although aluminum-containing cement has been widely adopted as an environmentally friendly construction material, the atomic-level understanding of its nanocrystalline nature remains unclear due to the diversity of chemical composition and structural defects in calcium aluminate silicate hydrate (C-A-S-H). In this study, we comprehensively characterize the atomic structures of C-A-S-H with Ca/Si ratios ranging from 1.3 to 1.9 and Al/Si ratios from 0.05 to 0.15. The composition, structural and mechanical features of C-A-S-H are accurately captured by molecular dynamics simulations of 1,600 distinct C-A-S-H structures constructed using an in-house automatic structure generation program, CASHgen. Our findings highlight the influence of Ca/Si and Al/Si ratios on key C-A-S-H characteristics, including the mean chain length (MCL), interlayer spacing, coordination number and elastic moduli. Specifically, the increase in the Ca/Si ratio at relatively low values will strengthen the interlayer connection, resulting in an improved modulus. While the continuous rise in the Ca/Si ratio after surpassing a critical threshold will lead to a shortening of the MCL, causing a decline in modulus. Besides, the increase in the Al/Si ratio promotes the growth in the MCL, thereby enhancing the modulus. This study provides valuable insights into the atomic-level behavior of cementitious C-A-S-H and other nanocrystalline materials, paving the way for the construction of their materials genome database and bottom-up optimization of their performance. **(Line 8, Page 1)**

Comment 2: The design of new cements that require less energy is also a common strategy for reducing the carbon footprint.

Response: Thank you very much for your valuable suggestion. We fully agree that designing new types of cement requiring less energy is indeed an essential and effective approach to reducing the carbon footprint. We have taken your advice into account and have revised the article accordingly to

clearly highlight this point. The revised content in the revised manuscript is as follows:

Common strategies for reducing the carbon footprint of cement production include improving the efficiency of cement plants, design of new cements that require less energy and partially substituting clinker with alternative materials, such as supplementary cementitious materials (SCMs) ^{6, 7, 8, 9, 10}.

(Line 32, Page 3)

Comment 3: The introduction is not logically structured. The authors attempt to include too much information, some of which is not directly related to the main objective of this study. For example, the main findings of this study do not directly address the hydration kinetics of SCMs-based cementitious materials or the underlying causes of prolonged setting times. It is unclear why these topics are included in the introduction.

Response: Thank you very much for your constructive feedback. We recognize that we did not focus as sharply as we should have on the main topic of the research. Your comments have highlighted this issue, and we have now removed the unrelated information from the introduction. The revised content can be found in the revised manuscript as follows:

These materials are typically aluminum-rich, leading to calcium-aluminate-silicate-hydrate (C-A-S-H) as the predominant hydration product in SCM-based cementitious systems. C-A-S-H holds significant potential for carbon sequestration and the immobilization of harmful substances. While numerous studies have explored the properties and hydration kinetics of SCMs-based cementitious materials^{11, 12, 13, 14, 15}, atomic-level research nonetheless remains limited. Advancing the understanding of C-A-S-H structure and its mechanical properties at the molecular scale is critical to optimizing the performance of sustainable cementitious materials. **(Line 37, Page 3)**

Comment 4: There are vague expressions throughout the manuscript. For instance, "but the combined influence of these factors remains challenging to evaluate." The combined influence of these factors on what?

Response: Thank you for your valuable feedback on our manuscript. We have carefully considered the comment regarding the vague expression, particularly the phrase "but the combined influence of these factors remains challenging to evaluate". In the original manuscript, our intention was to highlight that the effects of both the Ca/Si ratio and Al/Si ratio on the structural parameters are intertwined, making it difficult to evaluate each factor in isolation. For instance, an increase in the Al/Si ratio leads to an enhancement of bridging aluminum's connection to the silicate chains, which contributes to an increase in MCL. Conversely, an increase in the Ca/Si ratio results in the breaking of silicate chains and deepening disorder within the structure. These factors influence the structural parameters in a combined manner, which is why we described their effects as "integrated." Following

the reviewer's suggestion, we have revised the relevant section as follows:

Assessment of the validity of C-A-S-H atomic structures is nontrivial. The structural characteristics of the C-A-S-H, such as MCL, 2H/Si, are closely related to the Ca/Si and Al/Si ratios. Relying solely on Ca/(Si + Al) as a variable to analyze structural characteristics is inherently limited and fails to capture the nuanced changes in C-S-H structures resulting from aluminum incorporation. (Line 78, Page 5)

Comment 5: The authors should carefully review the content and determine what belongs in the Results section and what should be included in the Methods section. Excessive modeling details are currently presented in the Results section. Instead, the authors should focus on investigating the new findings and conclusions derived from the constructed CASH model.

Response: Thank you very much for your insightful comments. Based on your suggestion, we have carefully reconsidered the organization of the content. We moved the setting of aluminate protonation states from the Results section to the Methods:

After obtaining the required parameters, CASHgen assembles the blocks from atom groups, with the total number of blocks depending on the permissible charge deviation within the system. CASHgen automatically calculates the necessary number of Al atoms in the supercell to achieve the user-defined Al/Si ratio. The first task is to fill the supercell with blocks containing Al, continuously adding them until the total Al atom count is satisfied. During this process, the coordination and placement of Al atoms (such as in bridging sites or interlayer) are also considered. To address the complexity of aluminate protonation states while maintaining usability, CASHgen allows users to modify the protonation states according to their needs. By default, it pre-defines one-protonated Al(IV), two-protonated Al(V), and four-protonated Al(VI). (Line 446, Page 25)

Additionally, we removed the content of construction of C-A-S-H models from the Results section to the Methods:

4.2 Model construction

A total of 1600 initial supercells of C-A-S-H structures were generated for various Ca/Si ratios (1.3, 1.5, 1.7, and 1.9) and Al/Si ratios (0.05, 0.10, 0.15, with interlayer aluminum at 0.15), with 100 samples for each combination. Additionally, ²⁷Al MAS NMR experimental results have revealed the presence of Al(V) in the interlayer^{34, 53}. Therefore, an additional set of C-A-S-H structures at Al/Si = 0.15 for various Ca/Si ratios, with 100 samples for each ratio were also constructed. These structures are particularly noteworthy because 20% of the aluminum atoms are in the form of interlayer Al(V), representing 5% of the total aluminum content. (Line 490, Page 26)

Comment 6: The validation of the proposed CASH model is insufficient. The current validation

involves only a few previous studies and comparisons using several statistical indicators. The authors should synthesize representative CASH structures, conduct detailed experimental characterization (e.g., XRD, NMR), and provide a more comprehensive comparison.

Response: Thank you for your valuable suggestions. We appreciate your recommendation to conduct further experimental characterizations such as X-ray diffraction (XRD) and nuclear magnetic resonance (NMR). However, we would like to point out that previous studies have already conducted comprehensive experiments, including XRD, NMR and other methods, to characterize the C-A-S-H atomic structures^{1, 2, 3, 4, 5, 6}. These studies provide comprehensive experimental foundations for supporting the effectiveness of our CASH model, which was cited in our manuscript.

Additionally, the C-A-S-H models constructed in this work were based on the block model theory^{7, 8}, which has been widely accepted in the field of C-(A)-S-H model construction. In the figure S15, we perform a comparative analysis of the C-A-S-H model developed in this study with two well-established experimental works (C/S1.2-A/S0.10 and C/S1.5-A/S0.10^{1, 9}). Results demonstrate that the structure proposed herein exhibits a high degree of consistency with the experimentally obtained XRD data, particularly in reflecting the characteristics of the C-A-S-H phase. The minor discrepancies observed can be attributed to the presence of calcium hydroxide and other impurities, which was documented in the referenced experimental studies. Processing XRD experimental data can yield additional experimental values for C-A-S-H structural parameters, such as atomic bond lengths, the short-range order of the structure, and long-range characteristics, which can help us further validate the accuracy of the proposed C-A-S-H model. We added this content in the manuscript as follows:

We also compared the experimental and computational XRD results for C-A-S-H atomic structure (Figure S15). This analysis presents a high degree of consistency between the C-A-S-H atomic structure proposed herein and experimentally obtained XRD data, particularly in reflecting the characteristics of the C-A-S-H phase. (Line 254, Page 15)

To further validate the rationality of the proposed structure, Figure 4 (Section 2.3) in the manuscript compares the pair distribution function (PDF) and radial distribution function (RDF) data derived from total scattering experiments with the structural features of our proposed C-A-S-H model. This comparison confirms that our structural model accurately reflects the atomic-scale information of calcium-aluminum-silicate-hydrate (C-A-S-H) phases. Extensive supplementary characterizations are presented in Figures S9 to S14 of the Supporting Information to provide additional experimental evidence.

Additionally, the NMR experiments proposed by reviewers are critical for validating the rationality of the C-A-S-H structure, which we must prioritize. Through NMR experiments, we can specifically determine the mean chain length of the C-A-S-H model and the degree of atomic hydroxylation. We emphasized the NMR experimental content in Section 2.2, including detailed

validation of the model's mean chain length accuracy in Figure 2, and verification of the consistency between structural features obtained from NMR experiments and our proposed model in Figures 3(a)-(c).

Figure S1. Comparison of XRD experimental results and C-A-S-H model results. Experimental data for C/S1.2-A/S0.10 are from Wang J et al¹., and experimental data for C/S1.5-A/S0.10 are from Zhu X et al⁹.

Thus, we believe that our work is built on the solid foundation of the existing studies. Besides, our main aim is to contribute to advancing the understanding of the C-A-S-H atomic structures and providing insights from a broader range of model configurations by computational analysis, more detailed experimental characterization is beyond the scope of the work. We will follow the reviewer's suggestion to combine more insightful experimental work with the atomic simulation results in the near future work.

Comment 7: Section 2.3: Since the 160 samples were randomly selected, how can the authors ensure the results are representative? Please provide commentary on this.

Response: Thank you for your comment. The representativeness of the 160 models randomly selected is ensured through the consistency of a large amount of structural data and experimental comparisons. In Section 2.2, we compare the MCL value of the model in Figure 2, the 2H/Si ratio in Figure 3(a), the Si-OH/Si ratio in Figure 3(b) and the Ca-OH/Ca ratio in Figure 3(c) with experimental results. These comparisons show that these 160 C-A-S-H models generated by our CASHgen program can accurately reflect real C-A-S-H characteristics. Based on these results, we believe that these 160 C-A-S-H models can represent the corresponding atomic ratio in all C-A-S-H models with a reasonable and acceptable

range of fluctuations. Previous studies adjusted C-A-S-H models using Tobermorite, focusing on whether these indicators were reasonable, which aligns closely with our approach. Therefore, the 160 models accurately reflect the structural characteristics that should exist for the corresponding Ca/Si and Al/Si ratios while maintaining fluctuations within a reasonable range. This approach is more reliable than strictly adhering to specific numerical values.

Additionally, the representativeness of these 160 C-A-S-H models is also ensured by the strong ability of our CASHgen program to control the structural features of models within a reasonable range during their generation, which is reflected by the analyses of bond length through the radial distribution function (RDF) in Section 2.3, and the mechanical property tests in Section 2.4. We have to point out that these two structural and property analyses in past research were often performed based on a single model for a specific Ca/Si and Al/Si ratio, which is highly subjective. On the contrary, in this work, 16 different cases of CASH models were selected, and we chose 160 models because we wanted to provide more representative results for a specific atomic ratio, under the constraint of available computational power, rather than relying on the results of a single model. We averaged the results of numerous models with the same Ca/Si and Al/Si ratios, aiming to obtain a more real characteristics of C-A-S-H for a certain ratio and avoid overreliance on the particularities of a single model. For the subsequent analysis, averaging the parameters of 10 models for each specific ratio further avoids potential representativeness issues associated with a single model.

In summary, based on the results in Section 2.2, the C-A-S-H models provided by CASHgen reasonably represent the objects we are studying.

Comment 8 (Remarks on code availability): More detailed instructions about the code should be provided.

Response: Thank you very much for your valuable feedback. We have added the section “Code availability” and provided detailed instructions about the code in two files: "remade.md" and "docs.". The "remade.md" file includes key details about our program, the results of its installation and testing across different systems, and an explanation of the key parameters used in the execution text. The "docs" file offers a detailed description of the functionality of each module and the content it defines. We hope these materials will assist you in using our program effectively, and the revised content in the manuscript is as follows:

Code availability

The program CASHgen, developed for this paper, is available for download at <https://github.com/198y3j26/CASHgen>.

(Line 544, Page 29)

Reviewer #2

The manuscript “Nanocrystalline Structure of Calcium Aluminate Silicate Hydrate” (NCOMMS-25-06753) addresses a critical issue in cement chemistry, offering a comprehensive and innovative approach to investigating the nanocrystalline nature of calcium aluminate silicate hydrate (CASH). The authors have developed a user-friendly code for the automatic construction of CASH atomic structures. The manuscript considers various potential defects in CASH structures and a wide range of Ca/Si and Al/Si ratios. The atomic models of CASH accurately predict key structural parameters and mechanical properties of CASH, aligning closely with experimental results. The findings hold significant real-world implications for aluminate-containing cements and other nanocrystalline materials. Notably, the work has the potential to contribute to the creation of a cement structure database, which could aid in the development of machine learning-based force field potentials. Given the novelty, rigor, and potential impact of the study, I recommend publication in *Nature Communications* following the completion of the following minor revisions:

Comment 1: In Section 2.2, the authors summarize the results of some experimental work and perform a parameter surface fitting for the average chain length of CASH structures, based on experimental data collected at an Al/Si ratio below 0.2. Previous studies have typically considered the Ca/Si or Al/Si ratio as the sole variable affecting the average chain length. For CASH, it is difficult to accurately evaluate the influence of atomic content on chain formation or chain breaking. The authors' approach fills this gap and supports the accuracy of the structures generated by their proposed program. However, as high-alumina cements are increasingly recognized as green and sustainable materials, the author's fitting model currently only extends up to an Al/Si ratio of 0.2, with the applicability range stated as 0.15. Is there potential for further extension of this model?

Response: Thank you for your comment. In our attempt to fit the MCL surfaces for the calcium-silicon and aluminum-silicon ratios, we made every effort to extend the applicability of the fitted model, including for ratios such as Al/Si = 0.2 or higher, as suggested. However, after a comprehensive review of a wide range of experimental literature beyond the sources already cited in the manuscript, we identified several constraints that limit the further development of the model.

One primary limitation is the scarcity of MCL data for high-alumina systems. Additionally, synthesizing high-alumina C-A-S-H often poses challenges in achieving the desired calcium-to-silicon and aluminum-to-silicon ratios. Moreover, aluminum atoms predominantly occupy bridging sites, which leads to a sharp increase in the CASH model's MCL for high-alumina states. As a result, expanding the range of applicability could lead to overfitting, which we aimed to avoid. Therefore, we chose to limit the range of applicability to ensure the robustness and reliability of the model. Our reverse verification process, implemented within the program, confirms that this cautious approach is both meaningful and ensures the reliability of our model.

Comment 2: In Section 2.3, the authors introduce a parameter, “Ca²⁺/dimer,” which is related to the content in Figure 3(d). This parameter exhibits a strong linear relationship with the interlayer spacing. However, the authors do not provide a detailed explanation of “Ca²⁺/dimer.” It is only briefly mentioned in the fourth paragraph of Section 2.3, where it is suggested to be related to the number of calcium atoms in the interlayer. A clearer and more comprehensive explanation of this parameter is needed in the manuscript.

Response: Thank you for your suggestion. While “Ca²⁺” might generally be interpreted as referring to the calcium ions present in the interlayer, there is a distinction in this context. After hard minimization and prolonged equilibrium, we consider that the majority of the vacant bridging sites in the model will have interlayer Ca²⁺ ions nearby. Therefore, the effect of the Ca²⁺ ions still present in the interlayer on the interlayer spacing is significant. The term “dimer” refers to silicate dimers. Dividing the number of interlayer calcium atoms not near the vacant bridging sites by the number of silicate dimers allows for a more accurate assessment of the C-A-S-H model for different supercell sizes. Although the supercell format used in our study was consistently 4 × 4 × 2, this approach provides a more precise evaluation. The revised content could be found in the revised manuscript:

Assuming that, at equilibrium, the interlayer Ca²⁺ ions can occupy vacant bridging sites, the method involves first subtracting the number of vacancies from the Ca²⁺ ions quantity and then dividing by the number of silicate dimers. This approach has been proven effective in evaluating the impact of Ca²⁺ ions on the interlayer spacing. (Line 206, Page 13)

Comment 3: The research in this manuscript is well-supported by detailed data, and the Results and Methods sections provide references to supplementary materials, for example, in the Method section: “(See SI section 3).” However, excessive reference to supplementary materials can make the reading and retrieval process more cumbersome. I recommend that the authors revise this by providing specific reference numbers, such as Figure S1, instead of general supplementary information.

Response: Thank you for your recommendation. We have revised the supplementary information guidelines and made necessary modifications in the hope of achieving a more positive impact. The revised content as follows:

Therefore, the analysis of interlayer spacing changes can be focused on the interlayer distance. As shown in the RDF plot (Figure S10), the peak at 8.86 Å shifts to a greater distance with the increase in Ca/Si ratio, demonstrating that the change in interlayer spacing is primarily driven by the variation in the interlayer distance. (Line 275, Page 16)

The Young's modulus projection maps for specific Ca/Si and Al/Si ratios are summarized in Figure S17-S19. The impact of the Al/Si ratio on the Young's modulus of the structure can be observed in

Figure. 5(g). (Line 294, Page 18)

Consequently, the average quantities reported in this study were derived from the final 2 ns of the simulation (Figure S1). (Line 516, Page 27)

Comment 4: Figure 4(d) lacks a legend for the Si data. I suggest that the authors add annotations to some of the triangles in the figure to avoid ambiguity.

Response: Thank you for your valuable suggestion. We have added annotations to the relevant triangles in Figure 4(d) to clarify the Si data and avoid any ambiguity. The revised content could be found in the revised manuscript:

Figure 4. (a) The mean pair distribution function (PDF) of bulk C-A-S-H structure at the given Ca/Si or Al/Si ratios. The red circles highlight the Al-O peak at 1.8 \AA . The experimental PDFs are sourced from C.E. White et al.⁵⁸. (b) Mean Si-Si RDF at different Al/Si ratios (Ca/Si = 1.7); (c) Mean Si-Si RDF at different Ca/Si ratios (Al/Si = 0.15); (d) Annotation of Si-Si distance. The dark blue triangle represents the silicate tetrahedron.

(Line 257, Page 15)

Comment 5: In Figure 5(j-l), the authors correlate the parameters of the C-A-S-H structure with mechanical properties, which helps identify the key factors influencing mechanical performance. While previous studies have primarily focused on experimental analysis of the structures, simulations have often been limited by challenges such as supercell construction and computational costs. The approach presented in this manuscript, which analyzes large-scale data, fills a significant gap in the field. However, compared to the figure 5(j-k), the figure 5(l) lacks some structural data. Could the

authors clarify whether this still accurately supports the conclusions drawn?

Response: Thank you for your insightful comment. The 40 models containing interlayer aluminum were part of the earlier work in this paper, where we aimed to comprehensively consider the possible states of aluminum. However, for mechanical property analysis, particularly regarding the impact on the elastic constant C_{33} , it is crucial to accurately assess the influence of calcium ions. Since models with interlayer aluminum complicate this analysis, we chose to focus on the 120 models that do not contain interlayer aluminum. This allowed us to provide a more precise evaluation of the effect of calcium ions on the mechanical properties along the C-direction.

Comment 6: The link to the program at the end of Section 4.2 would be more appropriately placed under the Data/Code availability section, in line with standard formal requirements.

Response: Thank you for your suggestion. In accordance with standard guidelines, we have moved the link to the program to the "Code availability" section. The revised content is as follows:

Code availability

The program CASHgen, developed for this paper, is available for download at <https://github.com/l98y3j26/CASHgen>.

(Line 544, Page 29)

Reference

1. Wang J, *et al.* Effect of Ca/Si and Al/Si on micromechanical properties of C(-A)-S-H. *Cement and Concrete Research* **157**, 106811 (2022).
2. Avet F, Boehm-Courjault E, Scrivener K. Investigation of C-A-S-H composition, morphology and density in Limestone Calcined Clay Cement (LC3). *Cement and Concrete Research* **115**, 70-79 (2019).
3. Kapeluszna E, Kotwica Ł, Różycka A, Gołek Ł. Incorporation of Al in C-A-S-H gels with various Ca/Si and Al/Si ratio: Microstructural and structural characteristics with DTA/TG, XRD, FTIR and TEM analysis. *Construction and Building Materials* **155**, 643-653 (2017).
4. Lothenbach B, Jansen D, Yan Y, Schreiner J. Solubility and characterization of synthesized 11 Å Al-tobermorite. *Cement and Concrete Research* **159**, 106871 (2022).
5. Raúl F, Ruiz AI, Cuevas J. Formation of C-A-S-H phases from the interaction between concrete or cement and bentonite. *Clay Minerals* **51**, 223-235 (2016).
6. Puertas F, Palacios M, Manzano H, Dolado JS, Rico A, Rodríguez J. A model for the C-A-S-H gel formed in alkali-activated slag cements. *Journal of the European Ceramic Society* **31**, 2043-2056 (2011).
7. Pellenq RJM, *et al.* A realistic molecular model of cement hydrates. *Proceedings of the National Academy*

of Sciences **106**, 16102-16107 (2009).

8. Kunhi Mohamed A, Parker SC, Bowen P, Galmarini S. An atomistic building block description of C-S-H - Towards a realistic C-S-H model. *Cement and Concrete Research* **107**, 221-235 (2018).
9. Zhu X, *et al.* Nature of aluminates in C-A-S-H: A cryogenic stability insight, an extension of DNA-code rule, and a general structural-chemical formula. *Cement and Concrete Research* **167**, 107131 (2023).

Dear Reviewers,

Thank you very much for your hard work and comments on our manuscript “Nanocrystalline Structure of Calcium Aluminate Silicate Hydrate” (Manuscript ID: NCOMMS-25-06753B). After studying the comments carefully, we have made the modifications. The *Revised Manuscript* and the *Revised Manuscript with Track Changes* have been submitted. Here are our **point-to-point responses** to the reviewers’ comments.

Reviewer #1

Comment 1: The authors have adequately addressed most of my comments. I recommend the following minor revisions prior to acceptance in Nature Communications.

The discussion is quite straightforward—that is somewhat superficial. A more in-depth analysis should explore the real-world implications of these results for the application of cementitious materials.

The abstract has been greatly improved; however, the findings section focuses primarily on the modulus. While the modulus is important, this emphasis appears too narrow and may lead readers to believe that the manuscript only investigates the relationship between various structural characteristics and the modulus. This perception is insufficient for a paper published in Nature Communications and could limit its appeal to a broader audience. Therefore, the abstract should be further refined.

Response: We thank the reviewer for the thoughtful and constructive feedback. We appreciate the recognition that most of the previous concerns have been addressed and are grateful for the additional suggestions to strengthen the manuscript.

Regarding the discussion section, we agree that a more in-depth analysis of the broader implications would enhance the impact of the work. Accordingly, we have expanded the discussion (see pages 23) to more explicitly connect our atomistic insights to real-world applications. Specifically, we now highlight that by analyzing the mechanical properties of a large number of structural models, we demonstrate that the performance of C-A-S-H can be directly predicted from fundamental parameters such as Ca/Si ratio, Al/Si ratio, MCL, and atomic composition. This capability significantly broadens the potential for large-scale structural analysis and materials design.

Regarding the abstract, we acknowledge the reviewer’s concern that the emphasis on mechanical moduli may give a narrow impression of the study. Based on the reviewer's suggestions and the inspiration provided, we have changed the title to "High-throughput atomistic modeling of nanocrystalline structure and mechanics of calcium aluminate silicate hydrate" to include two important aspects, the nanocrystalline structure and mechanics, in this work. In response, we have revised the abstract to more clearly articulate the breadth of our analysis, which includes structural characterization (e.g., chain length, interlayer density, coordination states, RDF/PDF) and their

correlations with both mechanical and compositional features. The revised abstract now better reflects the multi-faceted scope of our high-throughput investigation and its broader relevance. The revised content in the revised manuscript is as follows:

Although aluminum-containing cements have gained attention as environmentally friendly construction materials, the nanocrystalline structure and mechanical behavior of their primary hydration product, calcium aluminate silicate hydrate (C-A-S-H), remain poorly understood due to its complex chemical composition and structural disorder. Here, we present a high-throughput atomistic modeling framework to systematically investigate the structural and mechanical properties of C-A-S-H across a broad range of Ca/Si (1.3–1.9) and Al/Si (0–0.15) ratios. The compositional, structural and mechanical features of C-A-S-H are accurately captured by molecular dynamics simulations of 1,600 distinct C-A-S-H structures constructed using our in-house automatic structure generation program, CASHgen. Our findings highlight the influence of Ca/Si and Al/Si ratios on key C-A-S-H characteristics, including the mean chain length (MCL), interlayer spacing, coordination number and elastic moduli. Specifically, C-A-S-H exhibits optimal mechanical performance at a Ca/Si ratio of approximately 1.5, while further increases in Ca/Si introduce disorder and reduce stiffness. In contrast, increasing the Al/Si ratio promotes chain polymerization, leading to longer MCLs and improved mechanical performance. These results provide atomic-scale insights into the structure-property relationships in C-A-S-H and offer design guidelines for high-performance, low-carbon cementitious materials.

Reviewer #2

Comment 1: I would like to commend the authors for their diligent and thorough revision of the manuscript. The concerns and suggestions raised in the initial review have been comprehensively addressed, resulting in a significantly improved and polished manuscript.

The inclusion of more detailed discussions and the strengthening of the conclusions enhance the overall impact and rigor of the study. Furthermore, the manuscript now has a clear structure and logical flow, making the content accessible and engaging for readers. The figures are well-organized and effectively complement the text.

In light of these improvements, I am pleased to recommend this manuscript for publication in Nature Communications, as it presents novel insights that will greatly contribute to the field of cementitious materials genome database.

Response: We sincerely thank the reviewer for their thoughtful feedback and generous recommendation. We are grateful for the valuable comments and suggestions that helped us significantly improve the clarity, structure, and scientific rigor of the manuscript. We are pleased that the revised version now meets the expectations of the journal and clearly conveys the broader implications of our work.